# The Effects of Agricultural Product Exports on Environmental Quality

**Sayed Saghaian** [1,*] , **Hosein Mohammadi** [2,*] **and Morteza Mohammadi** [3]

1 Department of Agricultural Economic, College of Agriculture, Food and Environment,
University of Kentucky, Lexington, KY 40536, USA

2 Department of Agricultural Economics, College of Agriculture, Ferdowsi University of Mashhad,
Mashhad 9177948978, Iran

3 Department of Economics, Faculty of Literature and Humanities, Hakim Sabzevari University,
Sabzevar 9617976487, Iran

* Correspondence: ssaghaian@uky.edu (S.S.); hoseinmohammadi@um.ac.ir (H.M.)

**Abstract:** Concerns about the environmental degradation of agricultural activities have increased with trade openness and globalization. In this study, the effects of agricultural product exports on environmental quality are investigated using panel data and instrumental variable regression models for 23 developed and 43 developing countries during 2002–2020. The results indicate that the expansion of agricultural product exports from developing countries has a detrimental effect on the environmental quality of these countries. Total agricultural exports increase pollution due to greenhouse gas emissions in developing countries, while they decrease the $N_2O$ emissions in developed countries. Moreover, raw agricultural exports have a positive and significant effect on agricultural pollution emissions in developing countries, while they have a negative and significant effect on $N_2O$ emissions in developed countries. In many developing countries, export development is an important policy objective, and agricultural exports are among the most important export sectors. Hence, policymakers need to consider the effects of agricultural product exports on the environment and increase farmers' awareness about the environmental consequences of agricultural activities. A better understanding of the environmental impacts of agricultural exports from developing countries is highly recommended.

**Keywords:** agricultural products; environment; sustainable exports; openness; pollution

## 1. Introduction

With rapid economic growth in many countries, especially in the developing world, concerns about pollution, particularly from increased production in the agriculture sector have increased [1]. Nitrogen and phosphorus are the most important nutrients for agricultural productivity [2]. Agricultural pollution caused by over usage of these fertilizers and other chemicals and pesticides has had a detrimental influence on human health as well as the environment [3,4]. Around 80% of total $N_2O$ emissions are related to agricultural activities, and the prerequisite for environmental quality improvement depends on the revision of agricultural activities in eight principal dimensions: water resources, soil erosion, nonpoint source pollution, pesticides, fertilizers, deforestation, population pressures, and biodiversity [5].

In the early 1990s, public attention to environmental impacts and pollution caused by agricultural activities increased [6]. The developed and developing countries discharge around 11.2% and 37.6% of the world's total agricultural pollution to the environment, respectively [7]. Changing land use and especially destroying forests in attempts to create new farmlands for agricultural purposes is one of the main reasons for increased greenhouse gas emissions and environmental degradation in the developing world [8].

According to [9], the agriculture sector is the largest consumer of water and the principal source of the nitrate pollution of ground and surface water, as well as the main source of ammonia pollution. It is a major contributor to the phosphate pollution of waterways and the release of major GHGs including $CH_4$ and $N_2O$ into the atmosphere.

The major agro-environmental problems fall into two categories. First, there are problems categorized at the global level, for example increasing atmospheric concentrations of the GHGs including carbon dioxide ($CO_2$) and $N_2O$ through deforestation and crop production [10]. The second set of problems is found in discrete locations of most countries, but at present has no influential impact at the global level. Examples are the salinization of lands and the gradual increase of nitrate fertilizer residues in ground and surface water [9].

As agricultural activities intensified, these problems emerged in the developed countries in the 1970s, but they became a central problem in some developing countries during the past decades. Most of the negative agricultural impacts on the environment can be prevented or reduced by a mix of technological change and policy [11]. Governmental programs, utilization of organic farming techniques, pest management, management of fertilizers, and an improved understanding of data on the agricultural chemical impacts on the environment are initiatives underway to reduce agricultural chemical impacts on the environment [12].

The pollution caused by agricultural activities and its reasons has been addressed in many studies [12,13]. During the last four decades, while agricultural lands were reduced by around 10% in the developed countries, their crop production doubled and their livestock production increased, but emissions decreased by around 7%. At the same time, in developing countries, agricultural lands expanded by 13%, crop production doubled, livestock production tripled, and total emissions increased by around 34% [14].

Economic growth and trade also affect environmental quality, and it is influenced by it. There are many studies on the relationships between economic growth, trade, and environmental quality [15–19]. In OECD countries, trade is found to benefit the environment, but in non-OECD countries, trade has detrimental effects on $SO_2$ and $CO_2$ emissions, although it does lower biological oxygen demand (BOD) emissions in these countries [20].

In [21], the authors examined the interaction between trade and $CO_2$ emissions for 49 high-emission countries and their results indicated that trade openness had both positive and negative effects on $CO_2$ emissions, but the impact varied in different groups of nations. Evidence in [22] supported the existence of the environmental Kuznets curve (EKC) hypothesis, but only in some transitional economies; the inverted u-shape curve relationship between economic growth and environmental degradation is known as the EKC hypothesis.

In [23], it was shown that total carbon dioxide emissions increased because of trade liberalization and that there is a shift in the structure of production toward most carbon-intensive sectors. The authors in [24] also emphasized that the food trade can have different effects on environmental pollution. They concluded that there is a need for a more comprehensive, integrated approach to estimate the global impacts of food trade on the environment.

Developing countries implemented policies that support the export of agricultural products [25] and moved toward more trade liberalization to gain benefits in their path toward economic growth and development. With more trade liberalization, the concerns about environmental issues increase because more trade could mean more production, and that has historically meant more pollution [26].

While there are many studies on the effects of economic growth or trade on environmental quality, there are very few studies on the effects of agricultural product trade on environmental pollution and quality, and this study tries to fill that gap. Due to the importance of the agricultural economic subsectors, especially in developing countries, this study focused on this important issue. We investigated the effects of agricultural product exports on environmental indices related to agricultural activities of $N_2O$ and $CH_4$ emissions. The contribution of this study is the investigation into the effects of exporting raw agricultural

products and total agricultural products on the environmental quality indices of $N_2O$ and $CH_4$ emissions. We used a panel data approach for a sample of developed and developing countries during 2002–2020. Moreover, since the implementation of agricultural policy in production and exports has a considerable impact on environmental degradation, the analysis of the EKC relationship in the agricultural sector was examined for both developed and developing countries. The results show that the expansion of agricultural product exports from developing countries has a detrimental effect on the environmental quality of these countries. The hypothesis of the research is:

**Hypotheses 1 (H1).** *The effect of agricultural product exports on environmental degradation is different in developed and developing countries.*

**2. Methodology**

*2.1. Method*

The environmental Kuznets curve (EKC) is based on the theoretical foundation of the current study. The EKC is a hypothesized relationship between various indicators of environmental degradation and economic growth. According to the EKC, in the early stages of economic growth, pollution emissions increase and environmental quality declines, but beyond some level of economic growth, the trend is reversed so that at high-income levels, economic growth leads to environmental improvement. This implies that environmental impacts or emissions per capita are an inverted u-shaped function of per capita income. The EKC is named after Simon Kuznets who proposed that income inequality first rises and then falls as economic development proceeds.

The relationship between agricultural exports and the environmental index can be written as follows [27]:

$$ENV_{it} = \beta_0 + \beta_1 EXP_{it} + \beta_2 GDP_{it} + \beta_3 GDP_{it}^2 + \beta_4 Z_{it} + U_{it} \tag{1}$$

where $ENV_{it}$ represents the environmental index of country $i$ in period $t$. $EXP_{it}$ is agricultural product exports of country $i$ in period $t$, $GDP_{it}$ and $GDP^2{}_{it}$ are gross domestic product ($GDP$) per capita and its squared value, respectively, $Z_{it}$ is a vector of control variables used in the literature, and $U_{it}$ is the error term. Our hypothesis is that $\beta_1$, or the effect of agricultural product exports on environmental degradation is positive. There are numerous variables, such as the openness of the economy and population density that can be replaced with $Z_{it}$.

Research in [28,29] used trade intensity which was obtained through the ratio of exports plus imports divided by GDP as an explanatory variable. In [28], the EKC was developed by using multiple versions of the above model and in [30,31], a quadratic functional form was used to examine the relationship between economic development and environmental indexes. According to the theories of environmental economics, GDP per capita could be used as a control variable and there is an inverted u-shape curve relationship between economic growth and environmental degradation; this is representative of the EKC hypothesis.

Equation (1) can be estimated with the panel data approach, but the estimations probably would lead to a biased estimator because of the endogeneity problem. Two main causes of biased estimations are, (i) a probable reverse causality between environmental degradation and agricultural product exports. In fact, good environmental conditions may lead to increased production and exports through increased agricultural productivity and profitability of land [32,33]. (ii) The export of raw agricultural products can be used as a substitute for a series of variables such as climate conditions, technology, and so on, and therefore $E(EXP_i.U_i) \neq 0$. To solve these problems, Equation (1) was estimated by the two-stage least squares (2SLS) estimator. This method requires the use of instrumental variables for the endogenous explanatory variables. Sargan–Hansen tests evaluate the instrument's independence from error term in the special case of linear instrumental variables with respect to over-identification restrictions [34].

According to the literature, two variables were used as instruments for raw agricultural products, including the ratio of land used for agriculture in a country to total land and agricultural machinery, tractors per 100 sq. km of land. These variables are positively correlated with agricultural exports because they are inputs of the agricultural production function. In addition, these variables are uncorrelated with the error term because they simply have an impact on the environment through agricultural production. Equation (2) is a panel data regression model with endogenous variables:

$$Y_{it} = \theta Z_{it} + \beta X_{it} + \mu_i + v_{it} \qquad \begin{array}{l} i = 1, 2, \dots, N \\ t = 1, 2, \dots, T \end{array} \qquad (2)$$

In Equation (2), $Z_{it}$ is a vector of endogenous variables and these variables are correlated with $v_{it}$. $X_{it}$ is a vector of exogenous variables, $\mu_i$ is the error due to lag, and $v_{it}$ is the error due to the time series in each of the sections. A variety of econometrics methods have been developed to best fit with an emphasis on $\mu_i$. The estimator of the random effects defines $\mu_i$ as a distribution of a random variable but disrupting behavior characterized. Therefore, if we assume $\mu_i$ is uncorrelated with the other variables, we can use the random effect model. The authors of [35] introduced a type of G2SLS method that is known as the random effect model.

### 2.2. Data Description and Model Variables

In this study, the instrumental variable regression model has been estimated using the panel data approach for the 2002–2020 periods for 66 countries, including 23 developed and 43 developing countries. The variables have been calculated and reported in the form of natural logarithms for better scaling. The classification of countries into two groups was carried out according to the Human Development Index (HDI) [36] in 2014. Countries with more than a 0.8 HDI were eligible as developed countries and countries with an HDI between 0.6–0.8 were considered developing countries. A description of research variables and their sources and units are explained in Table 1.

**Table 1.** Description of variables and their source.

| Variable | Description | Unit | Source |
|---|---|---|---|
| $N_2O$ emission | Nitrous oxide emitted during agricultural activities | Thousand metric tons of $CO_2$ equivalent | World bank |
| $CH_4$ emission | Methane emissions from livestock and other agricultural practices | Thousand metric tons of $CO_2$ equivalent | World bank |
| Total agricultural exports | Export value of agricultural products | 1000 US $ | FAO |
| Raw agricultural export | Agricultural raw material export | Current US $ | World bank |
| Per capita GDP($-1$) | Per capita gross domestic product with one lag | Constant 2010 US $ | World bank |
| Education | School enrollment, primary and secondary (gross) | Number | World bank |
| FDI | Foreign direct investment | Percent of GDP | FAO |
| Agriculture value added | Value added by the agricultural sector | Constant 2010 US $ | FAO |
| Agriculture employment | Percent of total employment | - | World bank |
| GDP(2) | Gross domestic product squared | - | World bank |
| Ln trade | Trade openness index (export + import)/GDP | Percent of GDP | World bank |
| Ln land | Total agricultural land (% of land area) | Percent of land area | World bank |
| Agricultural machinery | Tractors per 100 sq. km of arable land | - | World bank |

Instrumental variables such as agricultural land (% of land area) and agricultural machinery are taken from the World Development Indicators (WDI) of the World Bank. Environmental variables are represented by two main environmental indexes of agricultural $N_2O$ and $CH_4$ emissions that are closely related to agricultural activities [37], and their information is taken from the World Bank.

## 3. Results and Discussions

Before estimating the effects of explanatory variables on agricultural $N_2O$ and $CH_4$ emissions, some tests are necessary. First, to avoid any spurious regression problems, the Levin–Lin–Chu test is used for the stationary status of the variables. In Table 2, the results of Levin–Lin–Chu stationary tests for all variables are reported.

**Table 2.** The results of the Levin–Lin–Chu stationary test of variables.

| Variables | Developed Countries | Developing Countries |
|---|---|---|
| Ln N2O emission | −4.3 *** | −1.2 × 10 *** |
| Ln CH4 emission | −4.3 *** | −1.9 × 10 *** |
| Raw agriculture export | −5.3 *** | - |
| Total agricultural exports | −3.9 *** | −6.0 *** |
| Per capita GDP(−1) | −9.7 *** | −5.2 *** |
| Education | −5.2 *** | - |
| FDI | −4.8 *** | - |
| Agriculture value added | $−4.8 \times 10^4$ *** | - |
| Agriculture employment | −4.3 *** | - |
| GDP(2) | −5.4 *** | −5.1 *** |
| Ln trade | −8.6 *** | $−1.0 \times 10^6$ *** |
| Ln land | −5.0 × 10 *** | −5.9 *** |
| Agricultural machinery | −2.1 ** | - |

**, *** denote significance at 5% and 1% levels.

The null hypothesis in the Levin–Lin–Chu test is that all panels (each time series) contain a unit root. According to the results of Table 2, all variables are stationary. Another important test is the validation of instrumental variables. In Table 3, the validation of land used for agriculture and agricultural machinery as instrumental variables for agricultural export is considered.

**Table 3.** First and second-stage testing results of instrumental variables.

| Test Statistics | Developed Countries | | | | Developing Countries | | | |
|---|---|---|---|---|---|---|---|---|
| | Total Agricultural Export | | Raw Agricultural Export | | Total Agricultural Export | | Raw Agricultural Export | |
| | $CH_4$ | $N_2O$ | $CH_4$ | $N_2O$ | $CH_4$ | $N_2O$ | $CH_4$ | $N_2O$ |
| Partial $R^2$ | 0.06 | 0.06 | 0.08 | 0.08 | 0.11 | 0.11 | 0.07 | 0.07 |
| F | 10.9 *** | 10.9 *** | 15.1 *** | 15.1 *** | 35.9 *** | 35.9 *** | 22.3 *** | 22.3 *** |
| Sargan statistic | 0.03 | 3.5 * | 0.434 | 0.004 | 4.9 ** | 5.1 ** | 2.7 | 1.8 |

*, **, *** denote significance at 10%, 5% and 1% levels.

According to partial $R^2$ and F statistics as a relevance test in the first-stage equation, the interclass correlation among total agricultural export and instrument variables in developing countries is stronger than in developed countries. Sargan's statistics confirm the validation of the over-identification of all instruments as a subset of raw agricultural exports at a 1 percent error level in both groups of countries.

In other equations, the over-identification test is satisfied at the 5 and 10 percent levels. With respect to the validity of instrument test results reported in Table 3, all regression equations were estimated using the instrumental variables (IV) estimator.

For the analysis, all three models were estimated to achieve a comprehensive investigation of the relationship between agricultural exports and environmental quality. The first model was centered on total agricultural exports and emissions. The second model focused

on the raw agricultural export effects and emissions, and finally, in the third model, the model with the total agricultural exports was re-estimated by adding the trade openness as an explanatory variable in the regression equation. In addition, the comparison between developed and developing countries in each model was carried out. In all models, the logarithm of agricultural export (total agricultural exports and raw agricultural exports) has also been considered an endogenous variable. The instrumental variables are the natural logarithm of agricultural land (% of land area) and the natural logarithm of the quantity of agricultural machinery (tractors per 100 sq. km). The natural logarithm of agricultural emissions ($N_2O$ and $CH_4$) has been used as a dependent variable and the estimation results are reported in separate tables.

To achieve the appropriate estimation model among the approaches suggested by researchers for the panel data with endogenous variables, all possible models such as fixed effects and random effects (EC2SLS and G2SLS) were considered. To provide a general view of how total agricultural exports and raw agricultural exports could affect environmental quality, the estimated signs and significant levels of each regression equation are presented in Table S1 in Supplementary Materials.

The best model was selected according to the Hausman statistics test, reported in Table S2 in Supplementary Materials. The Hausman test related to the regression equations that explain why developed countries' pollution due to agricultural exports rejects the null hypothesis. That is, unobservable factors that might simultaneously affect the emissions and independent variables are fixed for each country.

Similar results are achieved for the regression equations in the developing country group, except for those with raw agricultural exports as an endogenous variable. Based on these results, there is a reason to believe that replacing raw agricultural exports with total agricultural exports does have some influence on emissions across developing countries. Hence, to achieve efficient estimations, one of the random effect approaches such as EC2SLS or G2SLS with respect to overall $R^2$ was used. It is noteworthy that both significant Hausman tests and their negative values confirm the fixed-effect approach; it considers country and time effects and provides appropriately reasonable results [38].

The estimation results of the effects of total agricultural exports on environmental quality are reported in separate columns for each country group in Table 4.

**Table 4.** The effects of total agricultural exports on environmental quality.

| Independent Variables (Exogenous) | Developed Countries | | Developing Countries | |
|---|---|---|---|---|
| | CH₄ (Fixed Effects) | N₂O (Fixed Effects) | CH₄ (Fixed Effects) | N₂O (Fixed Effects) |
| Total agricultural exports (US $) | −0.04 | −0.31 *** | 0.08 ** | 0.08 *** |
| | (0.04) | (0.10) | (0.04) | (0.03) |
| GDP per capita | 1.22 ** | −5.78 *** | 0.32 | 0.37 |
| | (0.62) | (1.68) | (0.26) | (0.24) |
| GDP per capita squared | −0.06 * | 0.32 *** | −0.03 ** | −0.03 * |
| | (0.03) | (0.09) | (0.02) | (0.02) |
| Agriculture value added | −0.02 | −0.10 ** | 0.17 *** | 0.16 *** |
| | (0.02) | (0.04) | (0.04) | (0.04) |
| Agriculture employment | 0.10 ** | −0.24 * | −0.00 | 0.00 |
| | (0.05) | (0.12) | (0.00) | (0.00) |
| FDI | 0.01 *** | 0.01 ** | 0.02 ** | 0.00 |
| | (0.00) | (0.01) | (0.01) | (0.00) |
| Education level | 0.03 | −0.02 | −0.01 ** | −0.01 *** |
| | (0.03) | (0.06) | (0.00) | (0.00) |
| Constant | 2.29 | 8.29 | 3.56 *** | 2.99 ** |
| | (3.27) | (6.21) | (1.28) | (1.16) |
| Number of observations | 405 | 405 | 765 | 765 |
| R² within | 0.43 | - | - | 0.30 |
| R² between | 0.29 | 0.64 | 0.54 | 0.67 |
| R² overall | 0.25 | 0.63 | 0.52 | 0.64 |

*, **, *** denote significance at 10%, 5% and 1% levels.

By comparing the results of the two groups of countries, it can be concluded that increasing total agricultural exports significantly contributed to the $N_2O$ emissions negatively in developed countries, while environmental quality worsens in developing countries with increases in total agricultural exports. This is usually due to the lack of resources and advanced technologies that mitigate harmful emissions in the agricultural production and exports of developing countries. In developing countries, the use of cheaper minerals such as coal and fossil fuels is a known contributor to environmental degradation and pollution. Those countries also suffer from poor infrastructure, including rail, road, and air networks. In developing countries, these effects were significant but positive for both $N_2O$ and $CH_4$ emissions. These results are consistent with [39].

The inverted u-shaped relationship between GDP per capita and $CH_4$ emissions is observed in the developed countries, but there is a u-shaped relationship between GDP per capita and $N_2O$ emission in this group of countries. Increases in exports of agricultural products contribute to increases in fertilizer and pesticide consumption within nations, which contributes to a variety of direct and indirect impacts on the environment [39]. In developing countries, the effect of squared GDP per capita on both $N_2O$ and $CH_4$ emissions is negative and significant, which indicates that economic growth could decrease pollution emissions in this group of countries.

The FDI variable is statistically significant for both country groups, but the estimated coefficient is small indicating that the impact is low. The authors of [40,41] also reached similar results. With a rise in value-added of agriculture, both $N_2O$ and $CH_4$ emissions increased in the developing countries, while the $N_2O$ emissions in the developed countries decreased. This is similar to the results of [42]. This result indicates that increased economic activity helps mitigate harmful emissions in developed countries but adversely affects developing countries. This result is also consistent with the previous results indicating a lack of climate-friendly technologies in the developing countries' production processes. Finally, education as a measure of a country's public awareness has a negative effect on both $N_2O$ and $CH_4$ emissions in developing countries. Hence, education has a positive externality, improving the environmental quality of developing countries.

The first-stage estimation results in Table S3 show the effects of instruments and other exogenous variables on the total export of agriculture. The results of the second-stage regression model with a focus on the effects of raw agricultural product exports on environmental quality are reported in Table 5.

According to these results, raw agricultural exports have a positive and significant effect on agricultural pollution emissions in developing countries, while it has a negative and significant effect on $N_2O$ emissions in developed countries. Therefore, with a rise in raw agricultural exports in developing countries, environmental pollution increases due to $CH_4$ and $N_2O$ emissions. These results are similar to previous results reported in Table 4. In the developed countries, there is an inverted u-shaped relationship between $CH_4$ emission and GDP per capita, while there is a u-shaped relationship between $N_2O$ emission and GDP per capita. The estimation results of raw agricultural exports with the instruments and other exogenous variables are reported in Table S4.

The results show that our hypothesis, that the effect of agricultural product exports on environmental degradation is different in developed and developing countries, cannot be rejected.

In the two previous stages, the effects of agricultural total exports and agricultural raw exports on environmental quality were investigated without considering the trade openness index. To provide more information about the effects of agricultural trade on environmental quality, the trade openness index is added to the model as an explanatory variable. Results provided in Table 6 show that raising total agricultural exports would increase the pollution due to greenhouse gas emissions in developing countries, while it would decrease the $N_2O$ emission in developed countries. Trade openness also increases $N_2O$ emissions in developed countries, while it decreases $CH_4$ emissions in developing countries. These results are opposite to the findings in [43] which reported a benign effect and harmful effect in high-income and middle to low-income countries, respectively. However, reference [44]

showed that the composition of export products and economic complexity promotes energy use and $CO_2$ emissions.

**Table 5.** The effects of raw agricultural exports on environmental quality.

| Independent Variables (Exogenous) | Developed Countries | | Developing Countries | |
|---|---|---|---|---|
| | $CH_4$ (Fixed Effects) | $N_2O$ (Fixed Effects) | $CH_4$ (G2SLS) | $N_2O$ (EC2SLS) |
| Raw agricultural exports (US $) | −0.02 | −0.27 *** | 0.09 ** | 0.13 *** |
| | (0.03) | (0.08) | (0.04) | (0.04) |
| GDP per capita | 1.46 ** | −3.74 ** | −0.04 | −0.02 |
| | (0.68) | (1.54) | (0.29) | (0.66) |
| GDP per capita squared | −0.07 ** | 0.21 *** | −0.01 | 0.00 |
| | (0.03) | (0.08) | (0.02) | (0.03) |
| Agriculture value added | −0.03 | −0.24 *** | 0.23 *** | 0.22 *** |
| | (0.02) | (0.05) | (0.04) | (0.06) |
| Agriculture employment | 0.12 *** | −0.18 ** | −0.00 | 0.00 |
| | (0.04) | (0.09) | (0.00) | (0.00) |
| FDI | 0.01 *** | 0.01 | 0.01 * | −0.00 |
| | (0.00) | (0.01) | (0.01) | (0.01) |
| Education level | 0.03 | −0.01 | −0.01 ** | −0.01 *** |
| | (0.02) | (0.06) | (0.00) | (0.00) |
| Constant | −15.10 *** | 37.12 *** | 2.42 * | 1.47 |
| | (2.49) | (8.02) | (1.30) | (3.23) |
| Number of observations | 405 | 405 | 765 | 765 |
| $R^2$ within | 0.40 | - | 0.02 | 0.17 |
| $R^2$ between | 0.21 | 0.80 | 0.54 | 0.60 |
| $R^2$ overall | 0.18 | 0.79 | 0.52 | 0.58 |

*, **, *** denote significance at 10%, 5% and 1% levels.

**Table 6.** The effects of total agricultural exports on environmental quality with trade openness.

| Independent Variables (Exogenous) | Developed Countries | | Developing Countries | |
|---|---|---|---|---|
| | $CH_4$ (Fixed Effects) | $N_2O$ (Fixed Effects) | $CH_4$ (Fixed Effects) | $N_2O$ (Fixed Effects) |
| Total agricultural exports (US $) | −0.03 | −0.31 *** | 0.10 ** | 0.09 *** |
| | (0.04) | (0.10) | (0.04) | (0.03) |
| GDP per capita | 1.22 ** | −5.70 *** | 0.37 | 0.39 |
| | (0.62) | (1.58) | (0.26) | (0.24) |
| GDP per capita squared | −0.06 * | 0.30 *** | −0.04 ** | −0.03 ** |
| | (0.02) | (0.08) | (0.02) | (0.02) |
| Trade openness | 0.03 | 0.39 *** | −0.12 *** | −0.05 |
| | (0.05) | (0.13) | (0.04) | (0.03) |
| Agriculture value added | −0.02 | −0.11 *** | 0.17 *** | 0.15 *** |
| | (0.02) | (0.04) | (0.05) | (0.04) |
| Agriculture employment | 0.10 ** | −0.20 * | 0.00 | 0.00 |
| | (0.04) | (0.11) | (0.00) | (0.00) |
| FDI | 0.01 *** | 0.01 ** | 0.02 *** | 0.00 |
| | (0.00) | (0.01) | (0.01) | (0.01) |
| Education level | 0.03 *** | −0.02 | −0.01 *** | −0.01 *** |
| | (0.02) | (0.06) | (0.00) | (0.00) |
| Constant | 2.29 | 41.19 *** | 3.76 *** | 3.09 *** |
| | (3.26) | (8.31) | (1.31) | (1.17) |
| Number of observations | 405 | 405 | 765 | 765 |
| $R^2$ within | 0.43 | - | - | 0.30 |
| $R^2$ between | 0.44 | 0.81 | 0.59 | 0.69 |
| $R^2$ overall | 0.39 | 0.80 | 0.57 | 0.66 |

*, **, *** denote significance at 10%, 5% and 1% levels.

Overall, there is evidence of the inverted u-shaped EKC relationship between $CH_4$ emission and economic growth in developed countries. These results indicate that agricultural exports cause increased emissions in developing countries and trade openness is harmful in developed countries regarding $N_2O$ emissions. Results in Table S5 also show the first-stage regression if trade openness is added to the model.

## 4. Conclusions

In the past two decades, economists have investigated the effects of trade openness on environmental quality and their results have mainly shown that there is a negative relationship between trade openness and environmental quality. However, few studies have addressed the relationship between different components of trade on environmental degradation. In this study, the relationship between agricultural exports and some environmental indices has been considered.

The results of the instrumental variables estimator for the developed and developing countries indicate that when the exports of agricultural raw materials in developed countries increase, the environmental quality has not been statistically affected, while in developing countries, environmental pollution increases mainly through rising $CH_4$ and $N_2O$ emissions. In developing countries, export development is an important policy objective and agricultural export is one of the most important export sectors [45,46].

With the expansion of international trade, environmental concerns have become a global problem. Agricultural trade also has indirect effects on the environment because it displaces farmers onto marginal lands, leading to deforestation and soil erosion. In many developing countries, the area devoted to agricultural exports increases. In some cases, the environmental effects of the transition to export agricultural products can be significant and harmful [47].

Some researchers argue that in the short run, agricultural trade has a positive effect on agricultural environmental pollution, while in the long run, it does not support the hypothesis that agricultural trade leads to environmental pollution [48].

Therefore, policymakers, especially in developing countries, should consider the effects of agricultural products export, especially raw agricultural exports, on environmental conditions. Increasing the awareness of farmers about the detrimental effects of using chemical inputs in agricultural activities is a suggestion in this regard.

Policymakers should also notice that production must be friendly to the environment with the adoption of new technology, higher productivity, and the regulation of exports of sustainable products [49].

Furthermore, developing related studies about the effects of each sector and subsector of the economy and the effect of the export and import of these subsectors on environmental quality is very important for targeting and controlling environmental degradation. Finally, although this article concluded that the export of raw agricultural products causes environmental degradation in developing countries, this result might not be true for all primary agricultural products. Therefore, further studies are needed to determine the effects of each product export and each sector export on environmental quality in each country.

**Supplementary Materials:** The following supporting information can be downloaded at: https://www.mdpi.com/article/10.3390/su142113857/s1, Table S1: The effect of agricultural products export and openness of trade on environmental quality in different models. Table S2: Selection process of suitable model. Table S3: The total agricultural exports effects on environmental quality-first stage results. Table S4: Raw agricultural exports effects on environmental quality-first stage results. Table S5: The total agricultural exports effects on environmental quality with openness of trade-first stage results. Table S6: Collinearity test results. Table S7: Heteroscedasticity test results.

**Author Contributions:** Conceptualization, S.S. and H.M.; methodology, S.S., H.M. and M.M.; software, H.M., M.M.; validation, S.S., H.M. and M.M.; formal analysis, H.M. and M.M.; investigation, S.S., H.M. and M.M.; resources, S.S. and H.M.; data curation, M.M. and H.M.; writing—original draft preparation, S.S. and H.M.; writing—review and editing, S.S.; visualization, H.M. and M.M.; supervision, S.S.; project administration, S.S. and H.M.; funding acquisition, S.S. All authors have read and agreed to the published version of the manuscript.

**Funding:** This research received no external funding.

**Institutional Review Board Statement:** Not applicable.

**Informed Consent Statement:** Not applicable.

**Data Availability Statement:** The data are available upon request.

**Acknowledgments:** The authors would like to thank the editors and the reviewers. Sayed Saghaian acknowledges the support from the United States Department of Agriculture, National Institute of Food and Agriculture, Hatch project No. KY004063, under accession number 7002927.

**Conflicts of Interest:** The authors declare no conflict of interest.

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
