# Peer review of "The Effects of Agricultural Product Exports on Environmental Quality"

_sustainability, doi:10.3390/su142113857_

Round 1

Reviewer 1 Report

Manuscript entitled “The effects of agricultural product exports on environmental quality” by Saghaian and team.

Abstract – Pl extend, it’s very short and add quantify data.  

Introduction – Ok, no further correction requires

M & M – Well define

Results and Discussions – L 176, 182, 184, 229, 265, 281, Pl see the references issues

Discussion should be more specific, with adding latest relevant references.

Conclusion – Well written

Overall, the manuscript has very information text, only discussion part should be improved before further process of this article.

Author Response

Responses to the Comments from Reviewer #1

Dear Reviewer,

Attached please find the revised manuscript. We addressed the reviewer's comments to the best of our ability and revised the manuscript accordingly. The useful comments helped improve the manuscript.

We would like to thank the reviewer for the helpful comments.

  1. Abstract – Pl extend, it’s very short and add quantify data. 

Response: Done. Thanks!

  1. Introduction – Ok, no further correction requires

Response: Thanks!

  1. M & M – Well define

Response: Thanks!

  1. Results and Discussions – L 176, 182, 184, 229, 265, 281, Pl see the references issues

 Response: Done. Thanks!

  1. Discussion should be more specific, with adding latest relevant references.

Response: Done. Thanks!

  1. Conclusion – Well written

Response: Thanks!

  1. Overall, the manuscript has very information text, only discussion part should be improved before further process of this article.

Response: Many Thanks!

Reviewer 2 Report

The article titled “ The Effects of Agricultural Product Exports on Environmental  Quality ” is very interesting, very well written, and informative. However, the author (s) must incorporate the following issues to improve the article.

1.     I could hardly understand the marginal contribution of this paper, as the authors review the literature in many aspect, which do not serve one research question. The authors must explicitly narrate the contribution/novelty of the study in one concrete paragraph in introduction section.

2.     The second main issue is the base or theoretical foundation of the study. On which particular theoretical or conceptual framework the topic/study is investigated. Please explain how equation 1 is constructed?

3.     Please provide a table of summary of recent literature on the topic in chronological order.

4.     The errors in line 182-183 as “According to the results of Error! Reference source not found.(2), all variables are stationary. Another important test is the validation of instrumental variables.  In Error! Reference source not found.(3)……may please be corrected.

5.     The results are meaningful but robustness or sensitivity analysis should be carried out to authenticate the results.

6.     Discussion of main results with contextualization i.e. consistency or contradiction with prior studies/events is missing. The author (s) needs to provide contextualization to the main findings.

7.     The policy implications should be drawn from obtained results and should be precisely linked with your study findings.

8.     I found some grammar and syntax errors. Please improve the overall write up of manuscript.

9.     The literature has ignored some recent studies. Please include the recent studies, for instance;

                           I.          https://doi.org/10.1080/09640568.2021.2008883

                         II.          https://doi.org/10.1016/j.energy.2021.122703

Author Response

Responses to the Comments from Reviewer #2

Dear reviewer,

Attached please find the revised manuscript. We addressed the reviewer's comments to the best of our ability and revised the manuscript accordingly. The useful comments helped improve the manuscript.

We would like to thank the reviewer for the helpful comments.

  1. I could hardly understand the marginal contribution of this paper, as the authors review the literature in many aspect, which do not serve one research question. The authors must explicitly narrate the contribution/novelty of the study in one concrete paragraph in introduction section.

Response: In lines 98-100 in the revised manuscript, we mention that: The contribution of this study is investigating the effects of total agricultural products export and raw agricultural products export on environmental quality indices of N2O and CH4 emissions. This issue has not addressed in the previous studies.

  1. The second main issue is the base or theoretical foundation of the study. On which particular theoretical or conceptual framework the topic/study is investigated. Please explain how equation 1 is constructed?

Response: The environmental Kuznets curve (EKC) is base or theoretical foundation of current study. EKC is a hypothesized relationship between various indicators of environmental degradation and indicators of economic growth. According to EKC, in the early stages of economic growth, pollution emissions increase and environmental quality declines, but beyond some level of economic growth, the trend reverses. So that at high income levels, economic growth leads to environmental improvement. This implies that environmental impacts or emissions per capita are an inverted U-shaped function of per capita income. The EKC is named after Simon Kuznets who proposed that income inequality first rises and then falls as economic development proceeds.

  1. The errors in line 182-183 as “According to the results of Error! Reference source not found.(2), all variables are stationary. Another important test is the validation of instrumental variables. In Error! Reference source not found.(3)……may please be corrected.

Response: Fixed. Thanks!

  1. The results are meaningful but robustness or sensitivity analysis should be carried out to authenticate the results.

Response: In this study, the relationship between the agricultural products exports and the quality of the environment is considered using the panel data regression method. Therefore, there is no need for sensitivity analysis because the t-statistics for each of the variables, which is obtained by dividing the coefficient of the variable by the standard deviation, shows that the variable in question is significant at the level of 1%, 5% or 10% and this issue can be the basis of analysis and reasoning.

  1. Discussion of main results with contextualization i.e., consistency or contradiction with prior studies/events is missing. The author (s) needs to provide contextualization to the main findings.

Response: Done. Thanks!

  1. The policy implications should be drawn from obtained results and should be precisely linked with your study findings.

Response: Done. Thanks!

  1. I found some grammar and syntax errors. Please improve the overall write up of manuscript.

Response: Fixed. Thanks!

  1. The literature has ignored some recent studies. Please include the recent studies, for instance;
  2. https://doi.org/10.1080/09640568.2021.2008883
  3. https://doi.org/10.1016/j.energy.2021.122703

Response: Done. Thanks!

Reviewer 3 Report

The article deals with a very interesting and important economic problem.

I have a few questions and comments:

- The authors used the Human Development Index. From which year?

- Explain why it was adopted: Education - unit Number?

‘In Error! Reference 176 source not found., the results of Levin- Lin- Chow…’ - ? (line 176 and 182, 184, 229 and further.

Author Response

Responses to the Comments from Reviewer #3

Dear Reviewer,

Attached please find the revised manuscript. We addressed the reviewer's comments to the best of our ability and revised the manuscript accordingly. The useful comments helped improve the manuscript.

We would like to thank the reviewer for the helpful comments.

questions and comments:

  1. The authors used the Human Development Index. From which year?

Response: The classification of countries into two groups of developed and developing countries was done according to the Human Development Index (HDI) from human development report in 2014. Countries with more than 0.8 HDI were eligible to developed countries and countries with HDI between 0.6-0.8 were considered as developing countries.

  1.  Explain why it was adopted: Education - unit Number?

Response: As mentioned in Table (1), education variable is number of school enrollment in primary and secondary schools.

  1. ‘In Error! Reference 176 source not found, the results of Levin- Lin- Chow…’ - ? (Line 176 and 182, 184, 229 and further).

Response: Fixed. Thanks!

Round 2

Reviewer 2 Report

I am satisfied and the authors have incorporated all comments. The article is publishable in this journal now.